# Determinants of the Implementation of Human Papillomavirus Vaccination in Zambia: Application of the Consolidated Framework for Implementation Research

**DOI:** 10.3390/vaccines12010032

**Published:** 2023-12-28

**Authors:** Mwansa Ketty Lubeya, Carla J. Chibwesha, Mulindi Mwanahamuntu, Moses Mukosha, Bellington Vwalika, Mary Kawonga

**Affiliations:** 1Department of Obstetrics and Gynaecology, School of Medicine, The University of Zambia, Lusaka 10101, Zambia; mulindim@gmail.com (M.M.); vwalikab@gmail.com (B.V.); 2Women and Newborn Hospital, University Teaching Hospitals, Nationalist Road, Ridgeway, Lusaka 10101, Zambia; 3School of Public Health, Faculty of Health Sciences, University of the Witwatersrand, Johannesburg 3193, South Africa; mukoshamoses@yahoo.com (M.M.); mary.kawonga@wits.ac.za (M.K.); 4Clinical HIV Research Unit, Helen Joseph Hospital, Johannesburg 2193, South Africa; carla_chibwesha@med.unc.edu; 5Department of Pharmacy, School of Health Sciences, University of Zambia, Lusaka 10101, Zambia; 6Department of Public Health Medicine, Charlotte Maxeke Johannesburg Academic Hospital, Johannesburg 2193, South Africa

**Keywords:** HPV vaccination, CFIR, implementation determinants, teachers, healthcare workers, COVID-19 vaccine, barriers, facilitators, parental consent, myths and misinformation

## Abstract

Cervical cancer can be prevented, primarily by the administration of the human papillomavirus (HPV) vaccine. Healthcare workers (HCWs) and teachers play important roles when schools are used for vaccine delivery; however, challenges exist. This study aimed to understand the barriers and facilitators to HPV vaccination that are perceived by HCWs and teachers. Guided by the consolidated framework for implementation research (CFIR), key informant interviews were conducted in Lusaka district between June 2021 and November 2021 using a semi-structured questionnaire. Recorded interviews were transcribed verbatim and imported into NVIVO 12 for data management and analysis. We coded transcripts inductively and deductively based on the adapted CFIR codebook. We reached saturation with 23 participants. We identified barriers and facilitators across the five CFIR domains. Facilitators included offering the HPV vaccine free of charge, HPV vaccine effectiveness, stakeholder engagement, and timely planning of the HPV vaccination. Barriers included vaccine mistrust due to its perceived novelty, low levels of parental knowledge, myths and misinformation about the vaccine, lack of parental consent to vaccinate daughters, lack of transport for vaccination outreach, lack of staff incentives, and inadequate sensitisation. Using the CFIR as a guiding framework, we have identified implementation barriers and facilitators to HPV vaccination among HCWs and teachers. Most of the identified barriers are modifiable, hence it is prudent that these are addressed for a high HPV vaccine uptake.

## 1. Introduction

The cervical cancer caused by high-risk human papillomavirus (HPV) types is the fourth most common cancer globally, with global estimates of 604,127 cases and 341,831 deaths for women [1]. However, most of these cancer cases occur in low- and middle-income countries (LMICs), where cervical cancer is the most common cause of cancer-related death. Furthermore, the risk of developing cervical cancer is increased by HIV co-infection by about six times compared with women living without HIV [1,2]. The burden of cervical cancer in sub-Saharan Africa (SSA) is worsened by the high prevalence of HIV/AIDS among women of reproductive age [3].

Zambia is a country with a very high cervical cancer age-standardised incidence of 65.5 per 100,000 women and a mortality of 43.4 per 100,000 women. [1,4]. Furthermore, the national prevalence of HIV in Zambia is about 12%; however, when disaggregated by sex, the prevalence in women is higher (14%) than in men (8%) [5]. The HIV epidemic further puts women at an increased risk of cervical cancer even when on antiretroviral therapy [6,7,8,9,10].

On the other hand, cervical cancer is one of the few preventable cancers, due to HPV vaccination, screening, and early treatment. However, in Zambia, HPV vaccine and cervical cancer screening coverage are low [11,12], just like in many SSA countries [2], leading to the late presentation of cervical cancer worsened by late diagnosis and delayed treatment at the only national chemoradiation centre [13].

HPV vaccination for adolescent girls intended for the primary prevention of cervical cancer was launched in 2019 in Zambia [12]. The vaccine coverage rates range from 31–76% for both doses 1 and 2, with some regions consistently reporting low figures [14]. The target group for HPV vaccination in Zambia is adolescent girls aged 14 years; the vaccination is delivered through static (health facilities) and outreach points (schools and within the communities) targeting in-school and out-of-school adolescent girls [12].

Multiple stakeholders are involved in HPV vaccine delivery, including teachers and healthcare providers. Teachers play a role in recruiting the in-school girls and sending information to parents, as well as in other activities such as ensuring order and filling out vaccination cards [12]. Primarily, healthcare professionals play a critical role in promoting HPV vaccination. Their duties range from planning, administering the vaccine, offering health education, entering data in the vaccination cards, and monitoring recipients for adverse events following immunisation. The aim is to maximise vaccination coverage and safeguard the adolescents’ health, however, this comes with challenges. The barriers to vaccinating HPV-vaccine-eligible girls span different aspects such as those related to the system and those related to the community [15,16,17].

Examples of barriers are: constraints on human resources, especially in outreach practice, as some members of staff must remain at the health facility to continue with other routine activities [18]; pockets of the eligible population that are hard to reach [19]; financial support to acquire logistics [15]; low levels of knowledge among healthcare workers, teachers, and parents of adolescents and belief in myths and misinformation [20]. Implementation facilitators include knowledgeable healthcare workers who can discuss the vaccine confidently and recommend it [21], stakeholder engagement, and adequate social mobilisation [15]. However, barriers and facilitators to HPV implementation are not well known in the Zambian context, where HPV vaccine coverage remains lower than the recommended threshold for herd immunity and there is high cervical-cancer-related incidence and mortality. Understanding the barriers is important for the development of appropriate strategies for this context and to enhance implementation success.

The successful implementation of HPV vaccination for adolescent girls is poised to increase coverage and reduce the incidence of HPV-related conditions such as cervical cancer in countries such as Zambia with high related morbidity and mortality. Therefore, this study aimed to understand the implementation determinants of HPV vaccination among teachers and healthcare workers in Lusaka, Zambia using the consolidated framework for implementation research as a guiding framework (CFIR) [22]. CFIR is one of the most widely used frameworks for exploring implementation determinants, especially in health-related research [23].

## 2. Materials and Methods

### 2.1. Study Design

This was a qualitative study with healthcare workers and teachers involved in the HPV vaccination program, as key informants.

### 2.2. Study Setting

The study was conducted within health facilities and schools that are located within the six sub-districts of Lusaka. Further details about the study setting and how the HPV vaccination program is implemented in Zambia are reported elsewhere [12].

### 2.3. Sampling

In the first stage, nine schools within the six sub-districts of Lusaka were sampled purposively if they were participating in the HPV vaccination program and six health facilities servicing the selected schools were sampled. In the second stage, we purposively sampled teachers and healthcare workers who were involved in the HPV vaccination program implementation. Not all teachers and healthcare workers found at these facilities participate in the HPV vaccination program. Participants were given information about the study and those who agreed to participate signed informed consent forms. Additional consent was obtained for interview audio recordings.

### 2.4. Data Collection

We developed an interview guide consisting of 50 questions, sub-questions, and a variable number of probes adapted from the CFIR website (www.cfir.org) to assess barriers and facilitators as perceived by teachers and healthcare workers. Demographic data about the participants were collected, and other data were based on the CFIR constructs. Some of the CFIR questions focused on the characteristics of the HPV vaccine, cost implications for the vaccination program, community response, stakeholder engagement, organisational incentives, and working relationships with colleagues. The CFIR is a determinant framework made up of five domains with 39 CFIR constructs and sub-constructs that reflect the variables most likely to influence the implementation of evidence-based interventions [22]. Recently, based on feedback from users, the CFIR has been updated to 48 constructs and 19 sub-constructs while maintaining the five domains [24].

We defined and contextualised our study according to the updated CFIR within the five domains as follows: the innovation domain included factors related to the HPV vaccine and HPV vaccination program, the inner setting domain included factors related to the organisational context (the health facilities and schools), the outer setting domain included adolescent girls, parents, and the community, the individuals domain included the characteristics of teachers and healthcare workers such as public health nurses, general nurses, and midwives, and the implementation process included processes for planning and executing the HPV vaccination program.

We developed and refined the interview guide using an iterative process. The interview guide was piloted using two interviews and minor edits were made to increase comprehension and make the guide more succinct, this continued throughout the data collection process.

Participants were recruited purposively based on their involvement in the HPV vaccination program. Key informant interviews took place in a private room within the school premises for teachers and within health facilities for healthcare workers. The interviews were conducted between June and November 2021; the specific timing and date of the interviews were dependent on the availability of the participants. Interviews were conducted in English and recorded using a voice recorder.

To ensure anonymity during data analysis, each participant was allocated a unique identifier. Data were collected until theoretical saturation was reached [25]. We used this process of rigorous data collection to assure dependability.

### 2.5. Data Management and Analysis

Trained research assistants transcribed all the interviews verbatim, whereas one of the co-authors checked for accuracy by listening to segments of the audio recording. NVIVO 12.0 was used to manage and analyse the collected data. We used thematic analysis to identify emerging themes as described by Braun and Clarke [26]. Some studies guided by the CFIR have also used a similar data analysis approach, considering that CFIR use is flexible and can be used according to context and study design [27,28,29].

Data were checked thoroughly during data collection and analysis for confirmability. The publicly available CFIR codebook based on the original CFIR was adapted and updated using the updated CFIR [24] and used to deductively code the data. One of the authors (MKL) coded initial transcripts, which were then reviewed by another co-author (MK) until consensus was reached to ensure the credibility of the data analysis process.

The standards for reporting qualitative research (SRQR) tool was used to guide the writing of this manuscript [30].

## 3. Results

We interviewed a total of 23 participants from the six sub-districts of Lusaka, of whom 12 were teachers. Twenty participants were female (Table 1).

The results are reported as facilitators and barriers for each theme (CFIR domain) and broken down into sub-themes, as shown in Table 2.

### 3.1. HPV Vaccine Characteristics Domain

The innovation in the context of this research is the HPV vaccine. Participants reported more facilitators than barriers regarding the HPV vaccine.

There were many ways in which the participants reported the characteristics of the HPV vaccine as facilitators to its implementation. Some of the perceived facilitators of the HPV vaccine included the prior demonstration project and the vaccine being given free of charge and including accurate information about its effectiveness.

#### 3.1.1. HPV Vaccine Trialability (HPV Demonstration Project)

Zambia had its demonstration programme for the HPV vaccine between 2013 and 2017, which was perceived as a success by the participants.

The pilot study was first done in 2013. We were told that we were to give those girls between the age of 9 and 14 but later were changed to those in grade 4. The first pilot we did was very successful as we didn’t have any complaints from mothers or the clients themselves. 009 Midwife

#### 3.1.2. HPV Vaccine Is Offered Free of Charge

Zambia is one of the countries benefiting from the Gavi alliance; it receives subsidised vaccines that are later given to eligible girls free of charge.

So, some of the parents come back to appreciate as they don’t pay for the medicine (HPV vaccine) 021 Teacher

#### 3.1.3. HPV Vaccine Evidence Base

Participants perceived that the known benefits of the HPV vaccine, such as prevention of cervical cancer, made them confident to implement the programme.

(…) because we are sure we can fight cervical cancer, it is very effective as we are giving the girls to help us fight cervical cancer. 018 Teacher

Regarding barriers associated with the HPV vaccine, participants reported some myths within the communities that were perceived as a barrier.

#### 3.1.4. Myths and Misinformation about the HPV Vaccine

There are deeply rooted myths about the HPV vaccine that prevent the community from accessing the HPV vaccine. Participants reported having some resistance from the community due to these myths. The most prominent was that girls who receive the HPV vaccine will be infertile and thus they would shun it for fear of not bearing children in the future.

Ah the vaccine at first was very difficult because some parents were saying “Why are you targeting our children from 9 years to 14–15 years? This means you want our children in the coming generation not to come and have, to come and bear children.” 002 Midwife

### 3.2. Outer Setting—Community

Within the outer setting, which includes the community and in which the inner setting exists, more barriers than facilitators were reported by the participants.

Collaborative efforts within the community were among the facilitators experienced.

#### 3.2.1. Partnerships and Connection

Participants reported that having partnerships within the community facilitated the HPV vaccination program, such as councillors who serve as gatekeepers for the communities.

The main stakeholders we work with are the ward councillors who are the owners of the village because whenever you have to go into the community you have to involve those. Then we also have the health neighbourhood committees in our units who help us with sensitisation issues. Then we also have the teachers in schools because we know that they can sensitise. 012 Public health nurse

There were some perceived barriers to HPV vaccination within the community which included low levels of parental knowledge, lack of parental consent to vaccinate daughters, belief in myths and misinformation and other large-scale occurrences such as the COVID-19 pandemic.

#### 3.2.2. Critical Incidents Such as the COVID-19 Pandemic

The COVID-19 pandemic was an unexpected global event that the entire globe battled with. Participants reported that the COVID-19 pandemic prevention measures, such as lockdowns and vaccination, had an impact on the HPV vaccination programme. Participants perceived the pandemic as a barrier to HPV vaccination as the myths around the COVID-19 vaccines had a negative effect on the HPV vaccine as well. Some parents felt that their daughters were being given the COVID-19 vaccine under the pretext that it was the HPV vaccine.

Because of this COVID-19, most of the parents are not interested in having their children vaccinated thinking we are giving them the COVID-19 vaccine (…) Some say you will become a zombie after some years, some say that they will never have children after vaccination whilst others say you will have blood clots. Those myths are still in the community. I remember before COVID-19, even if not all the children would get parental consent, the number was higher than when we started giving the COVID-19 Vaccine. 004 Midwife

#### 3.2.3. Local Attitudes in the Community

The low levels of knowledge on HPV and the HPV vaccine posed a challenge to the vaccination program. Participants reported that parents did not trust the HPV vaccine due to a lack of knowledge and hence they withheld consent for their daughters’ HPV vaccinations.

The other challenge is parental consent. We need to educate parents even as we educate their children. So, there is that challenge of them not having the knowledgeon the HPV vaccine. 017 Teacher

Some parents refuse the children from getting vaccinated as they don’t know its benefits so if only they were to be sensitised, and that would make things easier. 007 Teacher

### 3.3. Inner Setting Domain

Regarding the internal environment, which in this case was the health facility and the schools, some barriers and facilitators were reported. The reported facilitators included relational connections and access to information about HPV through workshops.

Some reported facilitators included strong relationships within teams and access to knowledge about the HPV vaccine.

#### 3.3.1. Relational Connections among Implementers

Participants reported that having relational connections with other workmates were seen to make the work easier, as there were consultations between old and new nurses.

We have good teamwork. If we don’t understand something we ask each other together with the new nurses. 004 Midwife

#### 3.3.2. Access to Knowledge and Information about the HPV Vaccine

Training sessions on HPV vaccination for teachers and healthcare workers were viewed as important in information dissemination to the implementors as well as vaccine users.

We had a workshop here where the doctors came, it was a one-day workshop. So, they came with some books and we were told how to go about it (…) they talked about the meaning of HPV; how someone can get it, and the dangers of it if one is not vaccinated, and they talked a lot. They also talked about parents with negative attitudes towards the vaccine and told us to talk to them about the benefits of their children receiving it. 018 Teacher

#### 3.3.3. Materials and Equipment

On the other hand, there were some barriers that were perceived in the inner setting, such as the limited availability of materials and funding. Some logistics were not in place, such as vehicles and fuel to ensure that outreach vaccination posts were accessed on time to maximise on the HPV vaccination delivery.

So, you find that there is only one vehicle that is supposed to take the team to schools and the same vehicle is also supposed to run other errands for the hospital. 009 Midwife

#### 3.3.4. Funding for HPV Vaccination

Community volunteers play a role in supporting the HPV vaccination program. The participants felt that having some funding set aside to provide incentives for these volunteers was important for the success of the program. Incentives could take the form of money or other attire for certain weather conditions.

For us to make sure that this programme is a success, we have to sensitise and for that to work we need the PA system, we need batteries for the megaphone it’s not just a matter of using the PA system, we need also need people to also go door to door. So, the foot soldiers (community volunteers) need to be given something as an incentive because, at the end of the day, they would want to eat because it won’t be effective sensitisation (…). 014 Registered nurse

### 3.4. Individuals Domain

This theme focuses on individuals with influence on the attitudes and behaviours of others regarding the HPV vaccine. There were some facilitators reported such as having the support of opinion leaders and other members of the community.

Within the individuals domain, participants reported that some facilitators of HPV vaccination included the involvement of opinion leaders and implementation leaders.

#### 3.4.1. Opinion Leaders

Opinion leaders were reported to play a role in the program implementation by encouraging their followers to take the vaccine. People often listen to these leaders and follow what they say. Support from opinion leaders in the vaccination process was viewed as a facilitator to the success of the program. These opinion leaders represented political, traditional, and religious circles:

We have the churches, the church leaders are the ones helping us to sensitise(…) even the marketeers, the chairpersons of these markets because once you go to them, they sensitise them then you find people coming to us. We also use traditional healers. Politicians are also very influential, so we would also go to the MPs [Member of parliament] to help us with sensitisation. So those are the stakeholders we have been using. 009 Midwife

#### 3.4.2. Implementation Team Members

Participants reported that individuals collaborating and supporting implementation were viewed as facilitators to the HPV vaccination. These offered support in different ways, ranging from sensitisation to data collection.

So, we also have champions in communities, we would say in each unit, we would choose a champion for HPV so that one will spearhead even the sensitisation, and registering of pupils, and that makes coordination easier when we collect data from them. We also have champions in schools. Without them, we can barely run the program as a good. 012 Public health nurse

#### 3.4.3. Characteristics Sub-Domain (Capability)

Some participants reported that having been trained made them more comfortable to participate in the HPV vaccination program.

I went for training which I completed I was at UTH [University Teaching Hospital] where I was offered a certificate in cervical cancer screening (…) I have also participated in sensitisation on the transmission of HPV in our catchment area. 002 Midwife

### 3.5. Implementation Process Domain

This domain represents the implementation process itself and includes strategies and activities used to implement HPV vaccination. Participants described facilitators as planning early, i.e., in advance of the actual vaccination dates to select the eligible girls and community sensitisation using various platforms. Additionally, the tailoring of strategies to mitigate some known pre-existing barriers, as well as active engagement with the parents and adolescent girls. were perceived as facilitators.

#### 3.5.1. Planning for HPV Vaccination

Participants reported that timely planning and sensitising communities using various platforms was a facilitator, as it made the actual delivery much more manageable.

In each school, before starting the programme… we (healthcare workers) would go to a particular school and give the teachers a task to count how many girl children were between the ages of 9 and 14 who were eligible for the vaccine. They would give us (healthcare workers) the number and we would also ask them to give us the date when they expected us to go and vaccinate because we wanted the teachers to also educate the girl child as they are very influential to these girls and their parents. 009 Midwife

#### 3.5.2. Tailoring Strategies to the Local Context

Tailoring implementation strategies to mitigate known barriers, such as a low level of knowledge about HPV, was perceived as a facilitator. This was the case for implementers as well as users.

Yes, that education can be given to the parents because what we used to do some time back, we had drama, so we would go out like at the marketplace where those people are found we would go there at the market, and we would have drama teaching people on what exactly would happen if someone had cervical cancer and we would take posters that shows exactly what the cervical cancer is and the normal cervix is so by doing that, people were educated on that one, and that’s what resulted for the parents to bring the children for vaccination. 002 Midwife

#### 3.5.3. Engaging HPV Vaccine Recipients

Participants reported that various strategies were used to attract end users, such as community sensitisation in public spaces about the HPV vaccine among the parents and the girls regardless of school status.

We use the schools and markets (…) to also cater for those that are not in school as everyone passes through the market. We also use markets during sensitisation. Mothers in the market are mostly there from 6 a.m. to around 20 h (8 p.m.), so they can also get the information and decide to bring their children. 004 Midwife.

### 3.6. Participants’ Suggestions

To have some local and home-grown solutions through stakeholder engagement, participants were asked to make suggestions on possible ways to improve the HPV vaccine coverage. Some suggestions on how best the program can be implemented were given by participants. The suggestions focused largely on the use of multimedia platforms and community gatherings such as churches to raise awareness within the outer setting. Other suggestions included sensitisation and equipping teachers with the correct information way before the actual vaccination dates.

Yes, the TV, and radio, and they can also do that by using mobile network carriers like MTN or Airtel to be sending out awareness texts. Then the church also must be involved by making announcements in church which helps some people make better decisions with such issues. Change is a process, until one day they all decide to know the benefits of the vaccine. So, we don’t have to wait for the program to start and then start sensitising, it has to be a continuous thing. 004 Midwife

The teachers also need to be sensitised because as teachers, they can explain better to their pupils, unlike the child only having basic information at the last minute which makes it difficult to explain themselves to the parents. 001 Teacher

## 4. Discussion

This study reported on determinants for the implementation of HPV vaccination in Zambia framed in the five CFIR domains. The identified facilitators included offering the vaccine free of charge, the HPV vaccine being readily available, and the inclusion of stakeholders, including healthcare workers and support staff, teachers, religious leaders, parents, adolescent girls, and media personnel in the different stages of implementation. The identified barriers across these domains included: mistrust around the HPV vaccine due to its perceived novelty, lack of information and education materials in local languages, low levels of parental knowledge, myths and misinformation about the HPV vaccine, lack of parental consent to vaccinate daughters, lack of transport for vaccination outreach purposes, lack of staff incentives, and inadequate sensitisation.

We found that giving the vaccine free of charge and it being readily available were perceived as facilitators of the vaccination program. In Zambia, the HPV vaccine has been given free of charge to eligible adolescent girls during both the demonstration project and the national rollout program supported by Gavi, the vaccine alliance [31]. However, this privilege is not cross-cutting for sub-Saharan Africa, as in some countries such as Nigeria, users must pay for the HPV vaccine, creating a barrier to uptake [32].

Similar to our findings, vaccine recipients have reported free vaccination as one of the main facilitators of HPV vaccination [33]. However, this facilitator is overshadowed by a myriad of barriers [17], leading to a low HPV vaccine uptake within the region [15]. Therefore, policy regulations should promote free and sustainable HPV vaccine provision for equitable access, especially in regions with high cervical cancer morbidity and mortality, to achieve the goals set by the WHO for cervical cancer elimination [34].

Including key stakeholders in the HPV vaccination implementation process was another reported facilitator in this study. The vaccination process requires cooperation and collaboration from multiple stakeholders for it to be successful [35]. For example, healthcare workers play a central role [21,36]; when schools are used for outreach, school staff are key [12]; political figures play the role of champions [37]; and community and religious leaders are trusted and when they approve of the vaccine, their followers are receptive [19]. Parents of adolescent girls should consent to the vaccination of their children, who are usually below the age of legal consent [12], even when most countries are implementing an opt-out approach [38]. On the other hand, adolescents themselves should be knowledgeable about the vaccine and have good attitudes towards it [39,40], hence it is necessary to improve their knowledge [41].

Therefore, when the stakeholders are supportive of HPV vaccination, uptake is enhanced. These findings are further supported by a systematic review of sub-Saharan health system constraints and facilitators, which revealed that in countries where there is multiple stakeholder engagement, the programs had better implementation outcomes [15].

Some barriers to HPV vaccination implementation were reported in this study. For example, there was a lack of trust in the vaccine, as it was perceived as a new and untested innovation; this was compounded by a lack of educational materials in local languages. Similarly, a study on ethnic minorities in the UK found that language and understanding of information and consent forms about HPV were difficult to understand for persons with low literacy, which was a barrier that impeded parental consent [42]. On the other hand, parental non-consent and worry about the novelty of the HPV vaccine were reported in other setups [43,44,45] (just like what was reported here) as hindrances to uptake.

In our recent study in this same setting, we found that 54% of parents of adolescent girls consented to vaccinating their daughters [12]. This is despite the WHO encouraging an opt-out parental consent approach or contextualising the consent process to country-specific requirements [38]. Some of the factors associated with parental consent for daughters included high knowledge levels about HPV vaccine, higher socioeconomic status, and parental HIV status (positive). These challenges are not unique to Zambia [46,47] and are encountered in many other settings [48,49,50].

Therefore, continued education and information sharing in easy-to-understand language to the parents of vaccine-eligible girls is key to increasing vaccine trust.

Financing for logistics during implementation was another identified barrier by our study participants, despite the vaccine being readily available. Examples included transportation to outreach points and a lack of incentives for community-based volunteers who are not on routine government salaries. This seems to be a similar situation to Mbale, Uganda where low financing was deemed as a barrier to service delivery in the absence of non-governmental organisations supporting the HPV vaccination program [18]. Therefore, national governments should be encouraged and compelled to increase national health financing, which will include sustainable funding for vaccination programs [51].

Our study participants reported inadequate social mobilisation and myths and misinformation in the implementation process of HPV vaccination. Critical information did not reach the intended groups, which could have deepened myths and misinformation, making users shy away from the vaccine altogether. Communication strategies on HPV vaccination that are tailored to specific stakeholder groups are effective in the African context [16], other low-income countries [52], and high-income countries [53,54,55] Effective communication helps in debunking and countering the myths and misinformation that have been rampant with many vaccines including the HPV vaccine [56].

## 5. Limitations

This study was conducted within Lusaka district to obtain insights into the HPV vaccination program. However, the results could be applied to other parts of the country and within the region due to their credibility and transferability.

## 6. Conclusions

We identified facilitators and barriers at the provider, interpersonal, and practice levels as perceived by healthcare workers and teachers in Zambia. Additionally, this study demonstrated the dynamic interplay between these factors by using the five CFIR domains. Therefore, context-relevant implementation strategies for mitigating the modifiable identified barriers and enhancing the facilitators are important for improving HPV vaccine coverage, which will protect future generations of women from cervical cancer and other HPV-related conditions. Additionally, a high HPV vaccine coverage entails the attainment of herd immunity and ultimately elimination [57,58,59]. Future research on identifying the implementation strategies that are effective in this setup is key [35,60].

## Figures and Tables

**Table 1 vaccines-12-00032-t001:** Characteristics of participants.

Sn	Age	Sex	Profession
001	35	Female	Teacher
002	50	Female	Midwife
003	48	Male	Teacher
004	49	Female	Midwife
005	54	Female	Teacher
006	44	Female	Teacher
007	42	Female	Teacher
008	39	Female	General nurse
009	60	Female	Midwife
010	41	Female	Public health nurse
011	43	Female	Teacher
012	36	Female	Public health nurse
013	25	Female	General nurse
014	30	Female	Registered nurse
015	47	Female	Midwife
016	57	Female	Teacher
017	54	Male	Teacher
018	45	Male	Teacher
019	46	Female	Teacher
020	28	Female	Teacher
021	22	Female	Teacher
022	37	Female	Public health nurse
023	35	Female	Midwife

**Table 2 vaccines-12-00032-t002:** Summary table of results: CFIR domains and implementation determinants.

Sn	Domain	Facilitators	Barriers
1.	Innovation characteristics	HPV vaccine effectiveness HPV vaccine given free of charge Ability to pilot the HPV vaccination before national scale-up	Innovation evidence base: community mistrust about the effectiveness of the HPV vaccine
2.	Outer setting	Stakeholder engagement	COVID-19 pandemic Myths and misconceptions Lack of parental consent
3.	Inner setting	Positive relationships with workmates Training of staff involved in implementing the HPV vaccination	Inadequate funding Lack of transport during implementation Lack of appropriate information, education, and communication materials
4.	Individual’s domain	Having opinion leaders support HPV vaccination Identifying HPV vaccination champions Individual competence in HPV vaccination	
5.	Implementation process domain	Planning early Education sessions for implementers and users Engaging HPV vaccine users	

## Data Availability

Data are available on reasonable request from the corresponding author as this work is part of an ongoing PhD.

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
