# Peer review of "Determinants of the Implementation of Human Papillomavirus Vaccination in Zambia: Application of the Consolidated Framework for Implementation Research"

_vaccines, 2023, doi:10.3390/vaccines12010032_

Round 1

Reviewer 1 Report

Comments and Suggestions for Authors

This study evaluated facilitators and barrier to HPV vaccine uptake in female adolescents in Zambia by interviewing healthcare workers and teachers using the consolidated for implementation for research framework. This is an important study to identify areas to improve vaccine uptake, preventing cervical cancer and deaths in a sub-Saharan African country. Below are my comments:

1.      Please provide more details with regards to the numbers of sampling units. Were samples taken from all 6 districts? How many schools and healthcare facilities were sampled out of number of available. How many HCW and teachers were in the selected sampling units?

2.      Line 140 of methods, I believe it should read no new ideas were obtained.

3.      Table 2: define IEC

4.      Line 175-176: what is demonstration project? Also “the vaccine being given free of charge including its effectiveness” does not make sense. Please revise. I think the authors mean the vaccine being given free of charge, and accurate information about its effectiveness.

5.      Section 3.1. is laid out in a confusing manner with no explanation. It was basically an outline rather than a narrative summary. The other sections were better in this respect. But in general, I found the outline format of the results section to be tedious and repetitive to read. I suggest revising with attention to streamlining and clarity of results.

Comments on the Quality of English Language

Some minor revisions regarding sentence structure and incomplete sentences

Author Response

Thank you very much for your feedback to improve this manuscript.

Reviewer 1

This study evaluated facilitators and barrier to HPV vaccine uptake in female adolescents in Zambia by interviewing healthcare workers and teachers using the consolidated for implementation for research framework. This is an important study to identify areas to improve vaccine uptake, preventing cervical cancer and deaths in a sub-Saharan African country. Below are my comments:

  1. Please provide more details with regards to the numbers of sampling units. Were samples taken from all 6 districts? How many schools and healthcare facilities were sampled out of number of available. How many HCW and teachers were in the selected sampling units?

Thank you, samples were taken from all six sub-districts, nine schools and six health facilities were included. Of the available teachers and healthcare workers within the sampling units, not all are involved in the HPV vaccination program, only a smaller proportion. For example, 1-2 within a facility with maybe 30-40 members of staff. The edits have been applied in the methods sections.

  1. Line 140 of methods, I believe it should read no new ideas were obtained.

Thank you, this has been corrected

  1. Table 2: define IEC

       Thank you, Information, Education, and Communication –TEXT has been updated.

  1. Line 175-176: what is a demonstration project? Also “the vaccine being given free of charge including its effectiveness” does not make sense. Please revise. I think the authors mean the vaccine being given free of charge, and accurate information about its effectiveness.

Thank you, corrected. A demonstration project is implementing the HPV vaccination program on a small scale before rolling out to the national level.

  1. Section 3.1. is laid out in a confusing manner with no explanation. It was basically an outline rather than a narrative summary. The other sections were better in this respect. But in general, I found the outline format of the results section to be tedious and repetitive to read. I suggest revising with attention to streamlining and clarity of results.

Thank you, section 3.1 has been edited and the whole results section has been streamlined.

Comments on the Quality of English Language

Some minor revisions regarding sentence structure and incomplete sentences

Thank you, this has been done.

Reviewer 2 Report

Comments and Suggestions for Authors

The authors report the results of a qualitative study on factors impacting on implementation of HPV vaccination. The study was conducted in Zambia in 2021; a total of 12 teachers and 11 healthcare workers (HCWs) were interviewed using a semi-structured questionnaire. Applying the Consolidated Framework for Implementation Research (CFIR),  facilitators and barriers were identified. These are in line with previous studies conducted in Sub-Saharan Africa, and the conclusions contain indications for implementation. However, some aspects can be improved. In particular:

-M&M: the teachers and HCWs to include in the study were "purposively sampled" (line 103) and the interviews were conducted in English. Please, explain the reason for using the English instead of the local language and whether this strategy can have selected the respondents and introduced some bias.

-M&M: the vast majority of the interviewed (20/23) were females; does this proportion reflect the actual gender distribution among teachers and HCWs?

-Results: besides the five domains summarized in Table 2, no details are given on the questions used during the interviews; please, include the total number of questions and some examples.

-Discussion (Limitations session): in lines 434-436, it is pointed out that "Although the CFIR framework offers a thorough  lens for ....., it may still need more study and validation before it can be directly applied to HPV vaccine characteristics.", but no data are reported in the Results to support this statement.

Minor comments:

-line 37, it should read "Cervical cancer, caused by high-risk human papillomavirus (HPV) types ..."

-line 40: please, add "..... deaths for women."

Comments on the Quality of English Language

Minor editing of English language required.

Author Response

Thank you very much for your comments to improve this manuscript.

Reviewer 2

The authors report the results of a qualitative study on factors impacting on implementation of HPV vaccination. The study was conducted in Zambia in 2021; a total of 12 teachers and 11 healthcare workers (HCWs) were interviewed using a semi-structured questionnaire. Applying the Consolidated Framework for Implementation Research (CFIR), facilitators and barriers were identified. These are in line with previous studies conducted in Sub-Saharan Africa, and the conclusions contain indications for implementation. However, some aspects can be improved. In particular:

-M&M: the teachers and HCWs to include in the study were "purposively sampled" (line 103) and the interviews were conducted in English. Please, explain the reason for using the English instead of the local language and whether this strategy can have selected the respondents and introduced some bias.

Thank you. English is the official language in Zambia, considering that the participants were trained professionals, communicating in English was appropriate hence no bias was anticipated.

-M&M: the vast majority of the interviewed (20/23) were females; does this proportion reflect the actual gender distribution among teachers and HCWs?

Thank you, this proportion reflects the gender distribution in the vaccination programmes

-Results: besides the five domains summarized in Table 2, no details are given on the questions used during the interviews; please, include the total number of questions and some examples.

Thank you. The questions in the interview guide were about 50 with variable number of probes. However, considering that we used a qualitative approach, the actual questions and order was guided by the responses given by the participants. The questions were adapted from the CFIR guide and are available at https://cfirguide.org/guide/app/#/

The methods section has been updated accordingly.

Discussion (Limitations session): in lines 434-436, it is pointed out that "Although the CFIR framework offers a thorough lens for ....., it may still need more study and validation before it can be directly applied to HPV vaccine characteristics.", but no data are reported in the Results to support this statement.

Thank you, this statement has been edited.

Minor comments:

-line 37, it should read "Cervical cancer, caused by high-risk human papillomavirus (HPV) types ..."

Thank you, corrected

-line 40: please, add "..... deaths for women."

Thank you, corrected

Comments on the Quality of English Language

Minor editing of English language required.

Thank you